# Potential Effects of Essential Oil from *Plinia cauliflora* (Mart.) Kausel on *Leishmania*: In Vivo, In Vitro, and In Silico Approaches

**DOI:** 10.3390/microorganisms12010207

**Published:** 2024-01-19

**Authors:** Vanderlan N. Holanda, Thaíse G. S. Brito, João R. S. de Oliveira, Rebeca X. da Cunha, Ana P. S. da Silva, Welson V. da Silva, Tiago F. S. Araújo, Josean F. Tavares, Sócrates G. dos Santos, Regina C. B. Q. Figueiredo, Vera L. M. Lima

**Affiliations:** 1Laboratório de Lipídios e Aplicação de Biomoléculas em Doenças Prevalentes e Negligenciadas, Departamento de Bioquímica, Centro de Biociências, Universidade Federal de Pernambuco, Avenida Professor Moraes Rego, 1235, Recife 50670-901, PE, Brazil; thaise.gabrielle@gmail.com (T.G.S.B.); joao.ricardhis@ufpe.br (J.R.S.d.O.); rebeca.cunha@ufpe.br (R.X.d.C.); ana.pssilva3@ufpe.br (A.P.S.d.S.); 2Laboratório de Biologia Celular de Patógenos, Instituto Aggeu Magalhães, Departamento de Microbiologia, Avenida Professor Moraes Rego, 1235, Recife 50670-901, PE, Brazil; welson1535@hotmail.com (W.V.d.S.); regina.bressan@fiocruz.br (R.C.B.Q.F.); 3Colegiado de Ciências Farmacêuticas, Universidade Federal do Vale do São Francisco, José de Sá Maniçoba, S/N, Petrolina 56304-917, PE, Brazil; tiago.fsaraujo@univasf.edu.br; 4Departamento de Ciências Farmacêuticas, Universidade Federal da Paraíba, Rua Tabelião Stanislau Eloy, 41, Castelo Branco III, João Pessoa 58033-455, PB, Brazil; josean@ltf.ufpb.br; 5Laboratório de Tecnologia Farmacêutica, Instituto de Pesquisa em Drogas e Medicamentos, Universidade Federal da Paraíba, Cidade Universitária, Campus I, Castelo Branco III, S/N, João Pessoa 58033-455, PB, Brazil; socratesgolzio@ltf.ufpb.br

**Keywords:** cutaneous leishmaniasis, *Plinia cauliflora*, essential oils, chemotherapy, phlogistic signs

## Abstract

In the search for new chemotherapeutic alternatives for cutaneous leishmaniasis (CL), essential oils are promising due to their diverse biological potential. In this study, we aimed to investigate the chemical composition and leishmanicidal and anti-inflammatory potential of the essential oil isolated from the leaves of *Plinia cauliflora* (PCEO). The chemical composition of PCEO showed β-cis-Caryophyllene (24.4%), epi-γ-Eudesmol (8%), 2-Naphthalenemethanol[decahydro-alpha] (8%), and trans-Calamenene (6.6%) as its major constituents. Our results showed that the PCEO has moderate cytotoxicity (CC_50_) of 137.4 and 143.7 μg/mL on mice peritoneal exudate cells (mPEC) and Vero cells, respectively. The PCEO was able to significantly decrease mPEC infection by *Leishmania amazonensis* and *Leishmania braziliensis*. The value of the inhibitory concentration (IC_50_) on amastigote forms was about 7.3 µg/mL (*L. amazonensis*) and 7.2 µg/mL (*L. braziliensis*). We showed that PCEO induced drastic ultrastructural changes in both species of *Leishmania* and had a high selectivity index (SI) > 18. The in silico ADMET analysis pointed out that PCEO can be used for the development of oral and/or topical formulation in the treatment of CL. In addition, we also demonstrated the in vivo anti-inflammatory effect, with a 95% reduction in paw edema and a decrease by at least 21.4% in migration immune cells in animals treated with 50 mg/kg of PCEO. Taken together, our results demonstrate that PCEO is a promising topical therapeutic agent against CL.

## 1. Introduction

Leishmaniasis is a complex group of neglected infectious diseases caused by protozoa of the genus *Leishmania*. These diseases present three well-established clinical forms: cutaneous leishmaniasis (CL), mucocutaneous leishmaniasis (MCL), and visceral leishmaniasis (VL) [1]. VL is the most severe form of leishmaniasis and can be fatal if not treated appropriately [2]. On the other hand, CL and MCL are the most common forms of the disease, causing social segregation due to the severity of the cutaneous and mucocutaneous lesions. Regardless of the clinical form, leishmaniasis is amongst the most devasting pathologies in more than 80 countries, especially those in tropical regions, resulting in high morbidity and mortality rates [3]. There are approximately 399 and 556 million individuals at risk of acquiring CL and VL, respectively. Every year, 900,000 to 1.3 million new cases and 20,000–30,000 deaths associated with leishmaniasis are reported in endemic countries [4]. The cutaneous forms of *Leishmania*, mainly caused by *L. braziliensis* and *L. amazonensis*, are the most prevalent clinical manifestations of leishmaniasis in Brazil, where the disease is present in all states, with an increased incidence in the last 20 years [5,6].

Pentavalent antimonials are the first-line drugs for the treatment of leishmaniasis. These drugs are outdated, require intravenous or intramuscular administration, and have severe side effects due to their high toxicity [7]. Furthermore, due to the strong inflammatory response induced by CL, the search for treatments that can combat the parasites and contribute to the balance of inflammatory response is still urgent [8,9]. The use of plants as therapeutic sources for the treatment of diseases has been a common practice since the beginning of human civilization [10]. Nowadays, the biological properties of plants used in traditional medicine continue to play an important role in the drug-discovery process [11].

Myrtaceae is one of the most important families of botanical species with ethnopharmacological relevance. A number of species of this family, such as *Myrciaria floribunda* [12], *Psidium guajava* [13], *Eugenia brejoensis* [14], *Eugenia uniflora* [15], *Eucalyptus globulus* [16], and *Plinia cauliflora,* have proven to have promising therapeutic properties, such as anti-inflammatory, antimicrobial, antidiarrheal [17]. *Plinia cauliflora* is one of the most important plants of Brazilian flora. The fruits of *P. cauliflora*, popularly known as Jabuticaba, have great economic and nutritional importance [18,19]. These effects were mainly attributed to the compounds of their secondary metabolism, which can be produced in different parts of the tree, such as fruits, flowers, and leaves [20]. The main formulations obtained from this botanical species are extracts, infusions, and essential oils [12]. The latter has drawn the attention of the scientific community due to the wide variety of chemical compounds and the possibility of biotechnological applications [21].

The investigation of the biomedical potential of *P. cauliflora* has been carried out over the last few years, including the production of formulations prepared from extracts or essential oils [22]. Different species of the Myrtaceae family, including *P. cauliflora*, present a wide variety of organic compounds that have biological activities, such as anti-diarrheal [23], antimicrobial [24], and antioxidant [25]. Previous studies have pointed out the anti-inflammatory potential of ethanolic extracts from leaves and branches [26] and the anti-promastigote (*Leishmania amazonensis*) activity of the essential oil [21,22,23,24,25,26,27]. Although promastigotes are not clinically important evolutionary forms, as they are absent in human infection, the products obtained from preparations with *P. cauliflora* samples can be useful in the in-depth investigation of the leishmanicidal and anti-inflammatory potential of this vegetable product, especially on species related to the development of CL. These properties can be attributed to the chemical diversity of bioactive compounds present in formulations obtained from *P. caulifora* [27], especially when considering samples obtained from regions and climates with different environmental stimuli.

In this study, we aimed to investigate the leishmanicidal effect of essential oil from *Plinia cauliflora* leaves (PCEO) on *Leishmania amazonensis* and *Leishmania braziliensis*, the causative species of CL in Brazil, as well as its inflammatory potential in Swiss albino mice, in order to provide better and less-toxic options of treatment for such a condition.

## 2. Materials and Methods

### 2.1. Plant Material

*Plinia caulifora* Kausel was collected in 2016 at the municipality of Belém de Maria in the State of Pernambuco, Brazil (8.6231° S, 35.8419° W) (Figure 1). The botanical voucher specimen was identified and deposited at the Herbarium of the Institute of Agronomy of Pernambuco under number 90798.

### 2.2. Isolation and Chemical Characterization of the Essential Oil from P. cauliflora (PCEO)

To obtain the essential oil, the leaves of *P. cauliflora* were harvested, cleaned, ground, and extracted by hydrodistillation in a modified Clevenger apparatus (400 g/L) for 4 h. The PCEO was collected, filtered, and weighed to calculate the total yield percentage (p/p) and then stored in the dark under refrigeration (5 °C) until use. The chemical identification of PCEO constituents was performed using a gas chromatograph coupled with a mass spectrometer (GC/MS, Shimadzu QP2010 Ultra, Kyoto, Japan) with an RTX-5MS capillary column (30 m length × 0.25 mm internal diameter, film thickness of 0.25 μm). Helium was used as low carrier gas (0.99 mL/min) with a constant pressure of 8.2 psi. The oven temperature was increased from 60 °C to 240 °C at a rate of 3 °C/min for 3 min before being increased to 240 °C for 3 min and held at 250 °C for 3 min. An aliquot of the PCEO samples (1 μL) was injected in split mode (20:1). The injector temperature was 250 °C. The GC/MS analysis was recorded (40–500 *m/z*) per 0.30 scan/s with total analysis time of 60 min. The identification of compounds was confirmed by comparing their mass spectra and retention times with standards available in the NIST08 and Pherobase libraries, as well as with the mass spectra from the literature [28]. For the experiments, a stock solution of PCEO (90 mg/mL) was prepared in pure Dimethylsulfoxide (DMSO, Sigma-Aldrich, St. Louis, MO, USA) and then diluted in culture medium at 1 mg/mL. This solution was again diluted in culture medium (work solution) at different concentrations for the experiments. The final concentration of DMSO never exceeded 1%, a concentration that is not toxic for the protozoa or mammalian cells.

### 2.3. Cytotoxicity in Mammalian Cells

Cytotoxicity analysis was performed in Vero cells (ATCC^®^ CCL-81™) and in Balb/c mice peritoneal exudate cells (mPEC) according to the protocol established by Medeiros et al. [29]. Briefly, mPEC and Vero cells were seeded into 96-well plates (10^5^ cells/well) containing 100 μL of RPMI 1640 medium (Sigma-Aldrich, USA) plus 10% inactivated FBS and allowed to be adhered for 3 h at 37 °C in 5% CO_2_ atmosphere. After this, non-adherent cells were removed by washing with RPMI medium. Subsequently, the cells were treated with different concentrations of PCEO (230.4–7.2 μg/mL). The cell viability was assayed using the Cell Titer-GloLuminescent Cell Viability Kit (Promega, Madison, WI, USA) according to the manufacturer’s instructions and analyzed in a GloMax microplate reader (Promega). The concentration toxic for 50% of cell population (CC_50_) was estimated by linear regression analysis using the software SPPS 8.0 in three independent experiments in triplicate.

### 2.4. Leishmanicidal Activity of PCEO on Leishmania amazonensis and Leishmania braziliensis

The leishmanicidal effect of PCEO was determined as described by Ramos et al. [1]. *Leishmania amazonensis* (LTB0016) and *Leishmania braziliensis* (LTB2903) (10^6^ cells/mL) were incubated in the absence or presence of different concentrations of PCEO (28.8 to 0.9 μg/mL) for 48 h at 26 °C. The cell growth was evaluated by cell counting in a Neubauer’s chamber. The concentration able to decrease promastigote growth by 50% (IC_50_) was estimated by linear regression analysis as described above. Each assay was performed in triplicate in three independent experiments. To determine the leishmanicidal effect of PCEO on intracellular amastigote forms, mPEC (10^6^ cells/mL) was seeded in a 24-well plate on coverslips. Non-adherent cells were removed, and the remaining cells were then infected with promastigotes of *L. amazonensis* or *L. braziliensis* at the rate of 15:1 parasite/cells for 14 h at 37 °C. After the infection time, the cells were washed with a fresh medium and treated or not with different concentrations of PCEO (14.62–3.65 µg/mL). After the treatment, the mPEC were fixed with methanol for 5 min and stained with Giemsa solution (1:20) in distilled water for 20 min. The coverslips were mounted on slides with permanent mount medium (Etellan^®^ Merck Millipore, Rahway, NJ, USA). The IC_50_ for amastigotes was determined as described for promastigotes, and the survival index was obtained by multiplying the percentage of infected mPEC by the mean number of amastigotes per infected cell. All the assays on amastigote forms were performed in duplicate in three independent experiments. The selectivity index (SI) was calculated as the ratio of CC_50_ for mammalian cells/IC_50_ for promastigote or amastigote forms.

### 2.5. Scanning Electron Microscopy

To identify putative morphological changes induced by PCEO treatment on the parasites, scanning electron microscopy analyses were performed according to the protocol reported by Aliança et al. [30]. For this, promastigote forms of *L. amazonensis* and *L. braziliensis* treated or not with IC_50_ and 2 × IC_50_ for 48 h were harvested via centrifugation at 1500× *g*, washed in 0.1 M phosphate buffer, pH 7.2, and fixed for 2 h at 4 °C in a solution containing 2.5% glutaraldehyde/4% paraformaldehyde in 0.1 M phosphate buffer, pH 7.2. After washing in the same buffer, the cells were post-fixed with 1% osmium tetroxide/0.8% potassium ferricyanide/5 mM CaCl_2_ in 0.1 M cacodylate buffer, pH 7.2, for 1 h in dark conditions. After post-fixation, the samples were washed in the same buffer, dehydrated in graded ethanol series, critical-point-dried with CO_2_, coated with a 20 nm-thick gold layer, and observed with a JEOL T-200 scanning electron microscope (Akishima, Japan).

### 2.6. In Silico ADMET Analysis

To predict the pharmacokinetic and druglike properties of the major compounds of PCEO, the following online platforms were used: Swiss ADME, Swiss Target Prediction (SIB., Lausanne, Switzerland) PROTOX-II Server (Charité University, Berlin, Germany), Molinspiration (Molinspiration Chemin informatics, Nova Ulica, Slovak Republic), and Osiris Property Explorer and Osiris Data Warrior (Actelion Pharmaceuticals Ltd., Allschwil, Switzerland).

### 2.7. Carrageenan-Induced Paw Edema

Adult Swiss mice were randomly divided into five groups, each group containing six animals. According to Winter et al. [31], animals were fasting for 6 h prior to the test. Each group received a different treatment: PCEO (50 mg/kg, 100 mg/kg, and 200 mg/kg), indomethacin (50 mg/kg), and a control group was treated with the control (saline 0.9%), all orally treated, via gavage. One hour after oral administration, edema was induced by injection of 15 µL of 2% carrageenan in the right leg subplantar region of each animal and 15 µL of saline in the left leg subplantar region of each animal. With the use of a digital caliper (Stainless Hardened, China), thickness (in mm) of the mice paws was measured at 0 h, 1 h, 2 h, 3 h, and 4 h after the injections were applied. Edema was measured as the difference between left paw and right paw.

### 2.8. Xylol-Induced Ear Edema

Adult Swiss mice were randomly divided into five groups, each group containing six animals. According to Rocha et al. [32], animals were fasting for 6 h prior to the test. Each group received a different treatment: PCEO (50 mg/kg, 100 mg/kg, and 200 mg/kg), indomethacin (50 mg/kg), and a control group was treated with the control (saline 0.9%), all orally treated, via gavage. One hour after oral administration, edema was induced via application of 20 µL of Xylol over each face of the right ear, and saline was applied to the left ear. With the use of a digital caliper (Stainless Hardened, China), thickness (in mm) of the mice ears was measured at 0 h and 2 h after Xylol was applied. Edema was measured as the difference between left ear and right ear.

### 2.9. Carrageenan-Induced Peritonitis

The test was carried out according to Souza and Ferreira [33]. Mice were randomly divided into five groups, each group containing six animals. Each group received a different treatment: there was PCEO (50 mg/kg, 100 mg/kg, and 200 mg/kg), dexamethasone was used as a positive control group (50 mg/kg in water), and negative control group was treated with 100 µL of the control (saline 0.9%), all via gavage p.o. After 1 h, 1% carrageenan in 0.9% NaCl solution was used to induce peritonitis. After 4 h, animals were euthanized by cervical dislocation, and subsequently, 3 mL of sterile saline solution (containing 5 UI/mL of heparin) was injected into peritoneal cavity. An incision was made to open the peritoneum and allow the exudate to be collected under aseptic conditions for subsequent total and differential white blood cells (WBC) count by optical microscopy (Coleman-N107, Coleman Equipamentos para Laboratório Com. e Imp. Ltd., Brazil). For total WBC count, 2 mL of the peritoneal liquid was mixed with 0.4 mL of Turk solution, and one aliquot was placed in the Neubauer chamber (with a depth of 0.100 mm and an area of 0.0025 mm^2^), and WBCs were counted in an optical microscope and expressed as number of cells × 10^6^/mL. Differential cell count was performed using 100× oil immersion lens in a cellular rubbing stained with fast panoptic kit (LABORCLIN, Brazil); one hundred cells were counted and differentiated as neutrophils, lymphocytes, and monocytes. The differentiated cells were expressed in absolute numbers of the WBC total count [34].

### 2.10. Statistical Analysis

SPSS 18.0 software (IBM Co., New York, NY, USA) was used for evaluation of CC_50_ and IC_50_ via linear regression. Significance was defined from the values of *p* < 0.05 using an ANOVA and Dunnett test in GraphPadPrism 5.0 (Graphpad, CA, USA).

## 3. Results and Discussion

The extraction of PCEO via hydrodistillation provided a higher yield of essential oil (7.33% *v*/*w*) when compared to reports of other species of the same family, such as *Myrciaria tenella*, *Calycorectes sellowianus* [35], and *Myrcia paivae* [36]. The chemical characterization of crude essential oils has a pivotal role in identifying the major components and understanding their possible mechanism of action. In addition, once the active constituent is known, it is possible to develop and prepare synthetic analogues with better-controlled properties, higher reproducibility, and economic viability [37]. Chemical characterization of PCEO (Table 1) revealed the presence of 52 constituents, corresponding to 96% of the total oil content identified in the mass spectrometer database (spectral and chromatographic characteristics are available in Appendix A). As another member of the Myrtaceae family, the GC-MS analysis of PCEO revealed the predominance of sesquiterpenes. The main components of PCEO were the sesquiterpene β- cis-Caryophyllene (24.4%), followed by epi-γ-Eudesmol (8%), 2-naphthalenemethanol [decahydro-alpha] (8%), and trans-calamenene (6.6%). Previous studies have also identified caryophyllene, its oxide, and isomers as the main active constituents of essential oils from plants of the Myrtaceae family that have medicinal properties [38,39]. Essential oils from the leaves of *Plinia edulis* showed caryophyllene oxide as the main compound (39.3%) [40]. Other species, such as *Plinia cordiflora* and *Plinia trunciflora*, also presented high concentrations of β-caryophyllene (15.9% and 8.2%, respectively) [41].

Eudesmol was also identified as a constituent of the volatile oils of *P. edulis*, *P. cordifolia*, and *P. truncatus* but in smaller quantities when compared with *P. cauliflora*, varying from 1.0 to 3.4% [41]. The chemical differences found between essential oils from different species of the *Plinia* genus, or even among plants of the same species, can be attributed to a diversity of factors, including the amount of irradiance, climatic variations, and methodology of extraction [64]. In this regard, we highlight nine components of the PCEO obtained from *P. cauliflora* from Atlantic Forest not yet mentioned by previous works for this species: 2-naphthalenemethanol, bicyclogermacrene, cyclohexanemethanol, aromadendrene, octatriene, cadinadiene, naphthalene, cyclohexene, and eucalyptol. On the other hand, caryophyllene and eudesmol appear to be the most common volatile compounds for *P. caulifora* and other members of the Myrtaceae family [21,27]. 

The applicability of natural compounds as therapeutic sources requires the knowledge of their pharmacokinetic properties as well as the investigation of their cytotoxic potential on mammalian cells [6]. The cytotoxic potential of PCEO was investigated on mice peritoneal exudate cells (mPEC) and Vero cells. Compared to the control cells, our results showed that PCEO was only able to significantly decrease the amount of ATP produced by both cell types at concentrations higher than 28.8 μg/mL. None of the tested concentrations were able to completely inhibit the ATP production by these cell types. The cytotoxic concentration for 50% of mPEC and Vero cells (CC_50_) was 137.4 and 143.7 μg/mL, respectively (Figure 2). 

According to Ríos et al. [65], EOs presenting CC_50_ values between 100 and 500 μg/mL are considered moderately toxic. However, the CC_50_ value of PCEO for mPEC was considerably lower than those reported in the literature for pentamidine and Amphotericin B, which are considered to have great cytotoxicity [3]. It has been reported that essential oils containing caryophyllene and its isomers, as major constituents, show low toxicity and present a marked anti-inflammatory activity [66]. Souza et al. [67] found that non-cytotoxic doses of essential oils from *Eugenia jambolana* and *Psidium widgrenianum* inhibited nitric oxide production in a late inflammatory reaction. Silva Mazutti et al. [68] highlighted the anti-inflammatory potential of the essential oil of *Eugenia dysenterica* (Myrtaceae) and correlated the reduction in the inflammatory process to NO inhibition. Because the infection of macrophage cells by *Leishmania* spp. causes a strong inflammatory response, leading to tissue destruction, the anti-inflammatory activity of caryophyllene and its derivatives on PCEO could help to mitigate the deleterious effect of the parasite on the site of cutaneous lesion. Further investigation is needed to investigate the immunomodulatory potential of PCEO on the course of infection with *Leishmania* spp.

The in vitro leishmanicidal activity of PCEO was evaluated on *Leishmania amazonensis* and *Leishmania braziliensis* (Figure 3). This essential oil was able to significantly inhibit, in a dose-dependent way, the growth of infective promastigotes from both parasite species at all concentrations tested. At a higher concentration of PCEO, the cell growth was inhibited by 85.47% and 88.35%, with IC_50_ values of 5.77 ± 0.96 and 5.60 ± 1.74 μg/mL for *L. amazonensis* (A) and *L. braziliensis* (B), respectively.

Amastigotes are the intracellular forms of *Leishmania* spp., replicating within parasitophorous vacuoles inside macrophages and other cells of the mononuclear system, leading to the destruction of parasitized cells [69]. The treatment of infected cells with PCEO was able to significantly inhibit the survival of intracellular parasites inside these cells (Figure 4). The IC_50_ values for amastigote were 7.31 ± 0.53 for *L. amazonensis* and 7.26 ± 0.12 µg/mL for *L. braziliensis* (Table 2). Although our results showed that amastigote was a little more resistant to PCEO treatment than promastigote, PCEO wasstrongly active for both evolutive forms of the parasite with IC_50_ < 10 µg/mL. Boyom et al. [70] and Silva et al. [21] reported that the hydrophobic characteristic of the essential oils allows them to cross the cell membranes and reach the intracellular amastigotes within the parasitophorous vacuole (PV). However, the effect of low pH found inside PV on the stability of essential oil compounds, as well as the need for drug metabolization by macrophages prior to being biologically activated, can also influence the activity of PCEO on amastigotes [19]. Interestingly, the effect of PCEO on promastigotes and amastigotes for both species was very similar, suggesting that the targets and/or pathways of PCEO activity are similar in both *L. amazonensis* and *L. braziliensis*. This is particularly important in the context of leishmaniasis since various studies have demonstrated that the efficacy of the current CL treatment is variable and strongly dependent on the parasite species associated with the infection [71,72].

Rodrigues et al. [73] showed that the essential oil of *Syzygium cumini* was effective against promastigote forms of *L. amazonensis*, with an IC_50_ value of 38.1 μg/mL. In addition, the essential oils from leaves of *Calyptranthes grandifolia*, *Calyptranthes tricona*, *Eugenia arenosa*, and *E. pyriformis* also presented a leishmanicidal effect on *L. amazonensis*, with an IC_50_ varying from 13.72 to 31.27 µg/mL. Compared to these previous studies, PCEO was more effective, with an IC_50_ of less than 10 µg/mL in both developmental forms of *L. amazonensis* and *L. braziliensis* (Table 2). Several authors have demonstrated that sesquiterpenes, such as caryophyllene and eudesmol, the main constituents of PCEO, are biologically active against *Leishmania* spp. Moreira et al. [74] demonstrated that the essential oil from *Casearia sylvestris* and *Melampodium divaricatum*, as well as its main constituent, E-caryophylene, was effective against *L. amazonensis*, with greater selectivity towards the amastigote form. Despite the anti-Leishmania potential of these compounds alone, there is still the possibility that these substances act synergistically, supporting the potent leishmanicidal effect of PCEO found in this study.

The effectiveness of PCEO on both evolutive forms of the parasite, as well as the cytotoxic activity of this essential oil towards mPEC and Vero, are summarized in Table 2. The selectivity index allows us to determine how much a given drug is effective against the parasite compared to the mammalian cells. The selectivity index values for both evolutive forms of the parasite were higher than 18, regardless of the mammalian cell type or leishmanial species tested. It is usually assumed that compounds with a SI ≥ 10 are considered promising to progress in the development of a new drug treatment [75].

In order to verify whether PCEO was able to disturb the plasma membrane structure of the parasites, scanning electron microscopy (SEM) was performed. Untreated cells were characterized by their spindle-shaped morphology and a smooth cell surface. A prominent flagellum emerged from the flagellar pocket, located at the anterior end of the parasite body (Figure 5A,B). Significant changes in the morphology and size of cells could be observed in PCEO-treated cells compared with the control (Figure 5C,D). Treated cells presented shortening and rounding of the cell body, blebbing and shrinking of the plasma membrane, and were lacking a flagellum. These alterations were more pronounced at twice the IC_50_ of PCEO (Figure 5E,F). At lower concentrations of PCEO, it was possible to observe an increased number of dividing cells, suggesting that besides being cytotoxic, the PCEO also has a cytostatic effect on both parasite species. The above-mentioned ultrastructural changes can be related to the loss of viability and cell death [76,77].

The conventional therapy for leishmaniasis, based on the use of parental antimony, causes severe cytotoxicity, has a high cost, and induces resistance. Alternative drugs, such as pentamidine, also have undesired side effects and require strict medical supervision [78]. Currently, Miltefosine is the only prescribed antileishmanial drug available for oral administration, which should be the preferential route for the treatment of leishmaniasis in endemic regions [79]. However, the oral administration of free miltefosine is toxic to the epithelial cells of the gastrointestinal tract and, if administered intravenously, this drug interacts with erythrocytes, causing hemolysis. The knowledge of pharmacokinetics properties (Adsorption, Distribution, Excretion, and Toxicity, ADMET) can be a valuable tool to identify new compounds that can be used in oral formulations for leishmaniasis treatment. Furthermore, it can help to predict the toxic potential of these compounds [80,81].

In this study, we used an in silico platform to predict the ADMET [82] properties of the major constituents of PCEO (Figure 6): beta-cis-caryophyllene (CAR), 10-epi-gamma-eudesmol (EUD), 2-naphthalenemethanol [deanhydro.alpha] (2-NAP), and trans-calamenene (CAL). Lipinsky’s rule of five (RO5) is widely accepted as a parameter to estimate the likeness of new compounds [83]. According to these rules, the compounds should present logP ≤ 5, molecular weight ≤ 500, number of hydrogen bond acceptors (nON) ≤ 10, number of hydrogen bond donors (nOHNH) ≤ 5, and number of rotatable bonds (nRotb) ≤ 10. Additionally, the hit compounds should not violate more than one rule [83].

All four major constituents of PCEO (CAR, EUD, 2-NAP, and CAL) satisfied the RO5, indicating that these compounds may have positive potential for oral treatment (Table 3). CAR and CAL were predicted to have low gastrointestinal (GI) absorption, whereas EUD and 2-NAP have a high probability of crossing the intestinal epithelium. None of them was a substrate for P-glycoprotein (P-gp). The P-gps belongs to ATP-binding cassette transport proteins and actively transports a wide variety of molecules across the cell membranes, playing a major role in multidrug resistance (MDR) [84]. According to Li et al. [84], a given compound can interact with P-gps in three different ways: as substrates, inhibitors, or modulators. Substrates of P-gp are not only subject to multidrug resistance (MDR) in chemotherapy, but they are also associated with poor pharmacokinetic profiles [85].

The information about compounds being a substrate or non-substrate of P-gp is very important to understanding the active efflux of compounds through biological transmembrane barriers [86]. The development of topical therapy for CL has been pursued due to its theoretical local potentiation, low cost, and easy administration [87]. The prediction of the skin permeability coefficient (Log Kp) for the transport of a compound through the mammalian epidermis is based on the principle that the more negative the Log Kp, the less skin-permeant the molecule. In this regard, our results showed that all the compounds presented good skin permeability, with values of LogKp of −4.44, −5.25, −4.77, and −3.90 cm/s for CAR, EUD, 2-NAP, and CAR, respectively. According to these criteria, the least skin-permeant was EUD, and the most was CAL.

Cytochrome P450 (CYP) enzymes, particularly CYP1A2, CYP2C9, CYP2C19, CYP2D6, and CYP3A4, are responsible for the bulk of the metabolism of known drugs in humans [88]. All the compounds tested were predicted to be an inhibitor of at least one CYP protein. CYP3A4 is known to be one of the most important P450 isoforms for drug metabolism, interacting with more than half of all clinically used drugs [89]. In this regard, CYP1A2 is considered a risk factor for hepatoxicity. None of the major compounds of PCEO was predicted to inhibit this protein. CYP2C9 was predicted to be inhibited by three of the four compounds tested (CAR, EUD, and 2-NAP). This latter protein has an important role in the metabolism of NSAIDs (Non-Steroidal Anti-Inflammatory Drugs) and weakly acidic molecules with a hydrogen bond acceptor [90]. Thus, the drug combination of CAR, EUD, or 2-NAP with NSAIDs should be considered with caution. Our predicted results in ProTox-II showed that none of the compounds presented features for tumorigenicity and skin irritability. Moreover, the predicted DL_50_ for a murine model of toxicity was considerably higher (>1000 mg/kg) for all the compounds, indicating that the major constituents of PCEO are safe and may serve as a potential leader compound for the development of new drugs against CL.

The PCEO showed a strong effect against paw edema induced by carrageenan and ear edema induced by xylol (Table 4). The lowest dose of PCEO resulted in a 64.2% reduction, and the highest dose resulted in 95% on the paw edema at the first hour of evaluation. Throughout the whole experiment, animals treated with PCEO presented lower edema than the control group (saline 0.9%). At a dose of 200 mg/kg, PCEO had the same effect as Indomethacin. At the final point of the paw edema test, the lowest dose increased the anti-inflammatory response to 75.2%, and the highest dose maintained a 95% reduction in paw edema. 

Carrageenan is a polysaccharide with elevated potential to induce an inflammatory process via prostaglandin release [91]. Thus, the reduction in carrageenan-induced edema indicates that PCEO might diminish inflammation by blocking the mechanisms of production and release of prostaglandins, such as prostaglandin E. Other essential oils from leaves of fruit plants also presented similar effects due to the presence of sesquiterpenes, such as *Hymenaea cangaceira* [92], *Eugenia stipitata* [82], and *Verbesina macrophylla* [93].

The essential oils from other plants from the Myrtaceae family also present anti-inflammatory responses, and most of their mechanisms create responses of antioxidant activity, reduction in cell migration, and inhibition of cytokines, nitric oxide, and other inflammatory mediators [94,95,96]. In addition, a study with extracts of *Plinia cauliflora* leaves and branches also showed reduced croton oil-induced ear edema [26]. This potential was associated with a reduction in nitric oxide and a reduction in leukocyte migration. Thus, our results pointed to the vast biological potential of *Plinia cauliflora,* in which several constituents of its secondary metabolism might play an important role. 

Xylol stimulates sensory neurons, inducing mastocytes and other inflammatory cells to trigger the inflammation process [97]. This mechanism is more refined than those found with carrageenan because the inflammation caused by carrageenan starts via prostaglandin signaling and takes steps to initiate cell migration, while Xylol quickly mediates the increase in cell migration and production of cytokines. All doses of PCEO showed the same effect in the ear edema test. Our results showed that PCEO has a higher anti-inflammatory effect compared with the pharmacological control, Indomethacin, with 92% ear edema reduction. Similar results were also observed with extracts of the same parts of *Plinia cauliflora*, although the presence of flavonoids and tannins was associated with the response.

These results indicate that PCEO is more sensible to reduce cell-induced inflammation, which is a strong mechanism to combat the inflammatory process such as those induced by *Leishmania* spp. infection. In addition, the results of the peritonitis test corroborate this hypothesis (Figure 7). Mice treated with 50 mg/kg had a low effect on cell migration, but the total cell count was 21.4% lower than the untreated group (control), which reinforces the idea that the sesquiterpenes present in PCEO reduced the cell migration and the inflammatory response triggered by the immune cells. There was no difference between results found at 100 mg/kg and 200 mg/kg doses. Our results showed a total of 42.8% reduction in white blood cell count. All the cell subtypes were present in lower numbers compared with the control, suggesting a decrease in cell migration in the mouse peritonitis-induced model treated with PCEO.

Thus, the essential oil of *Plinia cauliflora* might be a strong candidate for anti-inflammatory medicines. Although further investigation on the anti-inflammation signaling in PCEO-treated animals is needed, the possibility of reducing prostaglandins, and specifically cell migration and cytokine production, highlights the pharmacological potential of PCEO.

## 4. Conclusions

PCEO was highly selective against *L. amazonensis* and *L. braziliensis*, presenting the potential to combat the inflammatory response and characteristics of lesions caused by cutaneous leishmaniasis. ADMET analysis showed that the main constituents of this essential oil (CAR, EU, 2-NAP, and CAL) have great potential for the development of oral or topical formulations. The lack of infected animals and more treatments for other complications should be addressed in future investigations. Although the mechanism of action of PCEO and its isolated constituents remains to be determined, our results strongly suggest PCEO as a potential therapeutic candidate of natural origin and selective efficacy against the main etiological agents of cutaneous leishmaniasis. Furthermore, the treatment of phlogistic signs can be another pharmacological use of PCEO.

## Figures and Tables

**Figure 1 microorganisms-12-00207-f001:**
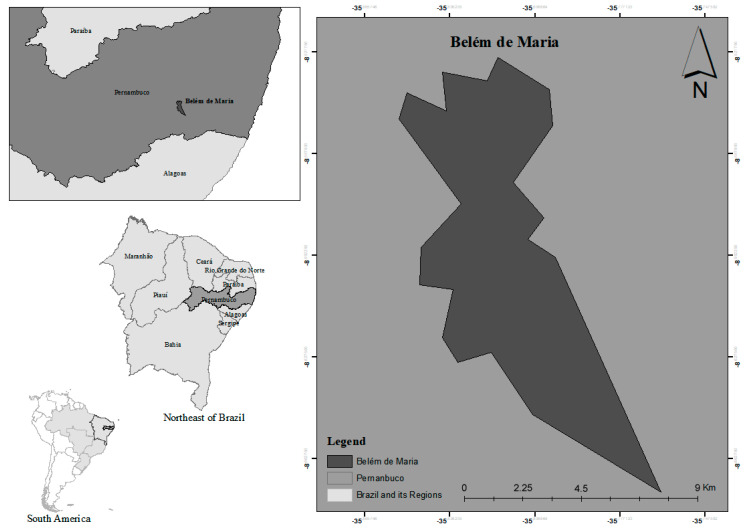
*Plinia caulifora* Kausel sample collection area.

**Figure 2 microorganisms-12-00207-f002:**
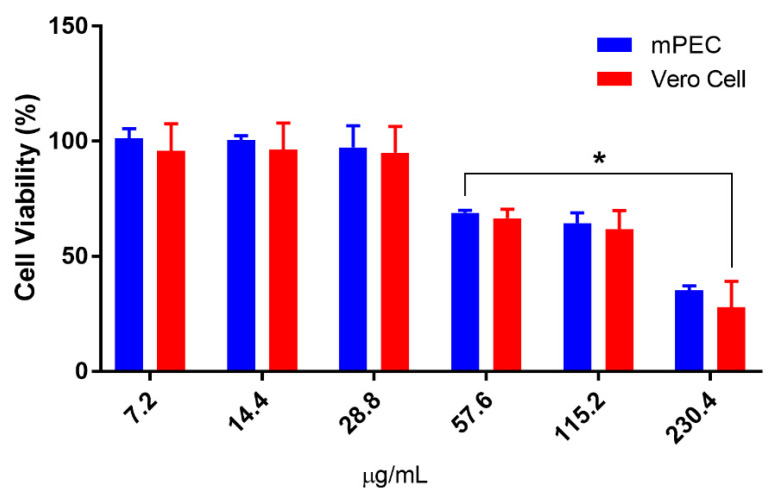
Effect of PCEO on the cell viability of mPEC Vero cells. The cell viability was assayed by measuring the intracellular ATP level using Cell-Title-Glo luminescent cell viability assay. Each cell type was analyzed in three independent experiments in triplicate, and data are shown as mean ± SD. * Significant difference compared to the control group (*p* < 0.05).

**Figure 3 microorganisms-12-00207-f003:**
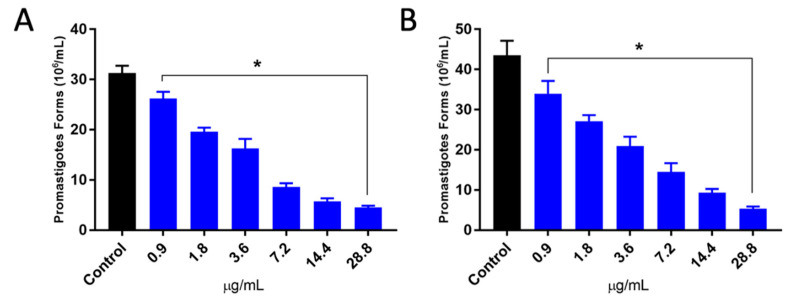
Effect of PCEO on promastigote forms of *Leishmania amazonensis* (**A**) and *Leishmania braziliensis* (**B**). The cell viability was assayed by measuring the intracellular ATP level using Cell-Title-Glo luminescent cell viability assay. Each cell type was analyzed in three independent experiments in triplicate, and data are shown as mean ± SD. * Significant difference compared to the control group (*p* < 0.05).

**Figure 4 microorganisms-12-00207-f004:**
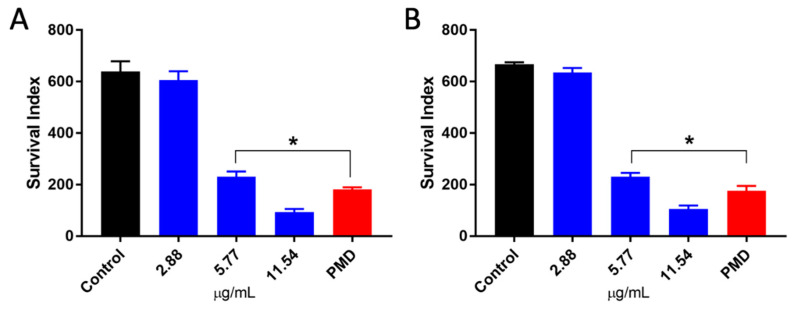
Effects of PCEO on *Leishmania amazonensis* and (**A**) *Leishmania braziliensis* intracellular amastigotes (**B**). Each bar represents the mean ± standard deviation of three independent experiments in triplicate. PMD: 30 µM pentamidine; * significant difference of each group compared to the control group (*p* ≤ 0.05).

**Figure 5 microorganisms-12-00207-f005:**
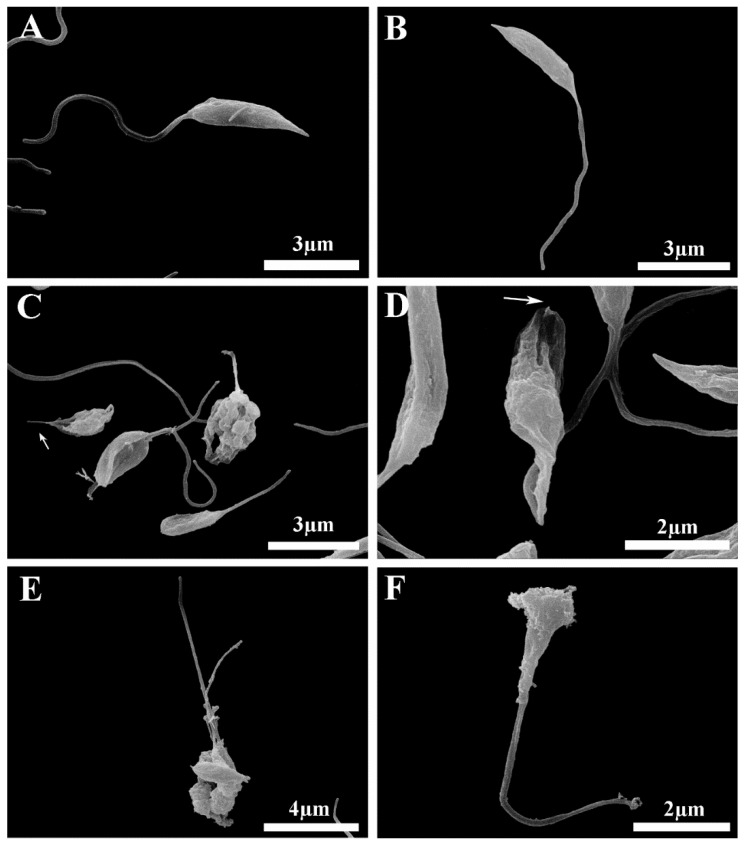
The effects of PCEO on the ultrastructure of *Leishmania braziliensis* (left column) and *Leishmania amazonensis* (right column), as observed by SEM. (**A**,**B**) Control cells; (**C**,**D**) low magnification of the promastigote culture treated with PCEO IC_50_, showing shortening and wrinkling of the cell body. (**E**,**F**) Promastigotes treated with twice IC_50_ PCEO presenting drastic morphological alterations, with shrinkage of the cell membrane. Some of the cells had short or absent flagella (white arrow).

**Figure 6 microorganisms-12-00207-f006:**
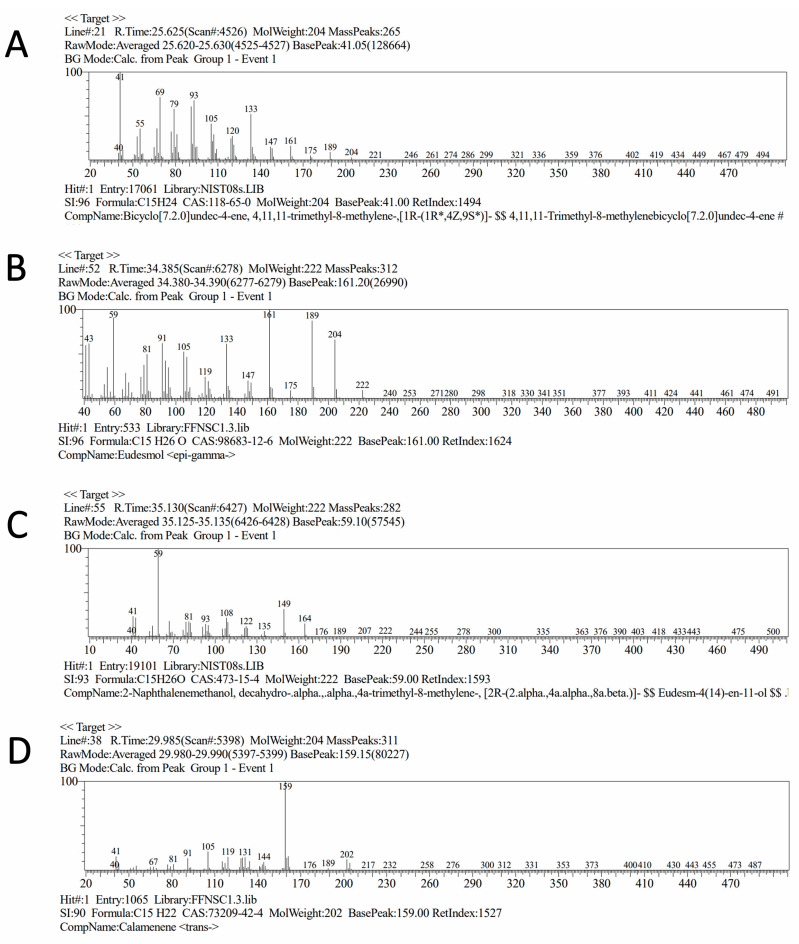
Mass spectrometry and chemical structures of major four constituintes of PCEO: β–cis–caryophyllene (**A**), epi–γ–eudesmol (**B**), 2-naphthalenemethanol [deanhydro.alpha] (**C**), and trans-calamenene (**D**).

**Figure 7 microorganisms-12-00207-f007:**
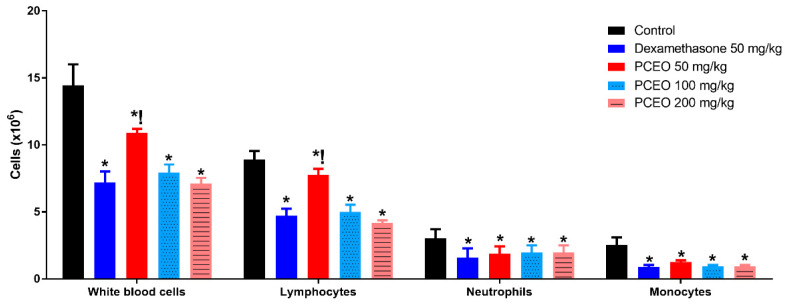
Effect of PCEO on inflammatory cells migration. PCEO: *Plinia cauliflora* essential oil from leaves. Data expressed as mean ± standard deviation. ANOVA and Dunnett test, *p* < 0.05, * vs. control; ! vs. Indomethacin.

**Table 1 microorganisms-12-00207-t001:** Main constituents of the essential oil of PCEO.

Compounds	IR ^1^	(%)	Reference
beta-cis-Caryophyllene	1494	24.04	[42]
epi-gamma-Eudesmol	1624	8.00	[43]
2-Naphthalenemethanol, decahydro-α,α,4a-trimethyl-8-methylene-, [2R-(2α,4aα,8aβ)]-	1593	8.00	[44]
trans-Calamenene	1527	6.69	[45]
2-Naphthalenemethanol	1598	5.84	[46]
Bicyclogermacrene	1497	5.46	[47]
Germacrene D	1480	4.07	[48]
Alfa-Copaene	1375	3.33	[49]
Cyclohexanemethanol	1522	2.60	[50]
Caryophyllene oxide	1507	2.52	[51]
β-Pinene	943	2.44	[52]
α-Caryophyllene	1579	1.92	[53]
Aromadendrene	1386	1.86	[54]
1,3,6-Octatriene	976	1.84	[55]
(-)-Spathulenol	1536	1.64	[56]
Cadinadiene	1452	1.63	[57]
Naphthalene [1,2,3,4,4a,7-hexahydro]	1536	1.47	[58]
Cyclohexene	377	1.29	[59]
tau-Muurolol	1580	1.06	[60]
Ledol	1530	1.05	[61]
α-Cubebene	1344	0.97	[62]
Eucalyptol	1032	0.88	[63]

^1^ IR (Retention Indices) of the compounds compared to the NIST08, Pherobase libraries, and Adams [28].

**Table 2 microorganisms-12-00207-t002:** Effect of PCEO on *Leishmania* spp. and mammalian cells.

			Promastigotes			Amastigotes		
Cell	CC_50_ ^1^ (μg/mL)	*Leishmania* Species	IC_50_ ^4^ (μg/mL)	IS_MØ_ ^5^	IS_VC_ ^6^	IC_50_ (μg/mL)	IS_MØ_	IS_VC_
mPEC	137.48 ± 4.6	*La* ^2^	5.77 ± 0.9	23.81	24.78	7.31 ± 0.5	18.80	19.66
Vero	143.73 ± 3.4	*Lb* ^3^	5.60 ± 1.7	24.53	25.53	7.26 ± 0.1	18.93	19.79

^1^ Cytotoxic concentration for 50% of mammalian cells; ^2^ *Leishmania amazonensis*; ^3^ *Leishmania braziliensis*; ^4^ Concentration able to inhibit the viability and growth of *Leishmania* spp. by 50%; ^5^ selectivity index of PCEO for mPEC; ^6^ selectivity index for Vero cells.

**Table 3 microorganisms-12-00207-t003:** ADMET ^1^ prediction of the toxicity parameters of the main four compounds from PCEO.

Parameter	CAR ^2^	EUD ^3^	2-NAPH ^4^	CAL ^5^
Lipinski Rules Violation	Yes	No	No	Yes
Physicochemical properties				
HBA ^6^ (≤10)	0	1	1	0
HBD ^7^ (≤5)	0	1	1	0
ClogP ^8^ (≤5)	3.29	3.19	3.11	3.19
MW ^9^ (≤500) g/mol	204.35	222.37	222.37	202.34
n-ROTB ^10^ (≤10)	0	1	1	1
Absorption				
BBB ^11^	No	Yes	Yes	No
HIA ^12^	Low	High	High	Low
P-GP ^13^ substrate	No	No	No	No
Skin permeability (cm/s)	−4.44	−5.25	−4.77	−3.90
Metabolism				
CYP450 2C9 inhibitor	Yes	No	Yes	No
CYP450 2D6 inhibitor	No	No	No	Yes
CYP450 2C19 inhibitor	Yes	No	No	No
CYP450 3A4 inhibitor	No	No	No	No
CYP450 1A2 inhibitor	No	No	No	No
Toxicity				
Mutagenic	No	No	No	No
Tumorigenic	No	No	No	No
Irritant	No	No	No	No
DL_50_ (mg/kg) ^14^	5000	5000	2000	1710

^1^ Absorption, Distribution, Metabolism, Excretion, and Toxicity; ^2^ CAR: beta- cis- Caryophyllene; ^3^ EUD: epi-gamma-Eudesmol; ^4^ 2-NPH: 2-Naphthalenemethanol [decahydro.alpha]; ^5^ CAL: trans-Calamenene; ^6^ number of hydrogen bond acceptors; ^7^ number of hydrogen donors; ^8^ logarithm of partition coefficient between octanol and water (lipophilicity); ^9^ molecular weight; ^10^ number of rotatable connections; ^11^ blood–brain barrier; ^12^ gastrointestinal absorption; ^13^ permeability glycoprotein P. metabolism; ^14^ DL_50_: lethal dose in 50%.

**Table 4 microorganisms-12-00207-t004:** Anti-inflammatory effects of PCEO on paw and ear edema.

Tests	Control	Indomethacin 50 mg/kg	PCEO ^1^		
			50 mg/kg	100 mg/kg	200 mg/kg
Paw edema					
0 h	0.13 ± 0.08	0.16 ± 0.08	0.08 ± 0.03	0.08 ± 0.05	0.10 ± 0.01
1 h	0.95 ± 0.19	0.05 ± 0.01 *	0.34 ± 0.04 *!	0.33 ± 0.07 *!	0.08 ± 0.06 *
2 h	1.05 ± 0.28	0.03 ± 0.02 *	0.34 ± 0.04 *!	0.25 ± 0.08 *!	0.05 ± 0.04 *
3 h	1.21 ± 0.26	0.03 ± 0.02 *	0.23 ± 0.08 *!	0.20 ± 0.07 *!	0.04 ± 0.01 *
4 h	1.01 ± 0.11	0.0 ± 0.01 *	0.25 ± 0.08 *!	0.15 ± 0.05 *!	0.05 ± 0.03 *
Ear edema					
0 h	0.04 ± 0.02	0.03 ± 0.01	0.04 ± 0.04	0.05 ± 0.04	0.03 ± 0.02
2 h	0.13 ± 0.01	0.52 ± 0.01 *	0.04 ± 0.01 *!	0.01 ± 0.01 *!	0.01 ± 0.01 *!

^1^ PCEO: *Plinia cauliflora* essential oil from leaves. Data expressed as mean ± standard deviation. ANOVA and Dunnett test, *p* < 0.05, * vs. control; ! vs. Indomethacin.

## Data Availability

Data are contained within the article and Appendix A.

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
