# Peer review of "Potential Effects of Essential Oil from Plinia cauliflora (Mart.) Kausel on Leishmania: In Vivo, In Vitro, and In Silico Approaches"

_microorganisms, 2024, doi:10.3390/microorganisms12010207_

Round 1

Reviewer 1 Report

Comments and Suggestions for Authors

In this paper, the authors present the evaluation of leishmanicidal and anti-inflammatory capacity of Plinica cauliflore essential oil. The research is interesting, organized and well-founded, the biological evaluation in vitro and in vivo the authors demonstrate the potential of the essential oil as a topical therapeutic agent against CL; however, some considerations for improving the work are suggested.

In section 3.4-Leishmanicidal activity, indicate the incubation temperatures.

In the results and discussion text, there are a large number of scientific names of microorganisms without italics.

References 48,49, 50 and 51 are the same

In Figure 2, adjust the “y” axis in A and B to the same scale for an easier and more accurate comparison.

The title suggests that the leishmanicidal evaluation was carried out in vivo and this was not the case, the leishmanicidal evaluation was only in vitro and in sillico; the anti-inflammatory evaluation was in vivo; I suggest correcting the title to make it clear and not create false expectations in the reader.

According to the results, I would like to know why the authors did not include an in vivo leishmanicidal evaluation with a LC model.

Author Response

Reviewer 1

Reviewer: In section 3.4-Leishmanicidal activity, indicate the incubation temperatures.

Our answer: Thank you for your kindly reminder. We already write the temperatures at this section, being 26°C for promastigotas and 37°C for amastigotas. Both additions are highlighted in yellow.

Reviewer: In the results and discussion text, there are a large number of scientific names of microorganisms without italic.

Our answer: Thank you for your kindly reminder. We already revised the entire manuscript and correct all scientific names for the italic format.

Reviewer: References 48,49, 50 and 51 are the same

Our answer: Thank you for your kindly reminder. We already revised all the references and correct what was incorrect placed. All the changes are highlighted in yellow.

Reviewer: In Figure 2, adjust the "Y" axis in A and B to the same scale for an easier and more accurate comparison.

Our answer: We understand your concern regarding the comparisons, but those figures are meant to show the potential of PCEO against the types of Leishmania rather than compare the activity on both species.

Reviewer: The title suggests that the leishmanicidal evaluation was carried out in vivo and this was not the case, the leishmanicidal evaluation was only in vitro and in sillico; the anti-inflammatory evaluation was in vivo; I suggest correcting the title to make it clear and not create false expectations in the reader.

Our answer: We are sorry for that. We have changed the title, so it can match better our aim and analysis.

Reviewer: According to the results, I would like to know why the authors did not include an in vivo leishmanicidal evaluation with a LC model.

Our answer: Thank you for kindly suggestion. Indeed, it will be great if we did an in vivo leishmanicidal evaluation, but due to laboratory and research logistic we are not able to perform this test. Also, it was not the purpose of the present work to do an evaluation at that extent, which does not invalidate the present findings and completeness of this work.

Reviewer 2 Report

Comments and Suggestions for Authors

In this research article entitled "Leishmanicidal Activities of Essential Oil from Plinia cauliflora (Mart.) Kausel: in vivo, in vitro, and in silico approaches", the authors investigate the chemical composition and the leishmanicidal and anti-inflammatory potential of the essential oil isolated from the leaves of Plinia cauliflora (PCEO). Quantitatively, enough experiments have been carried out and the results and discussion are in most cases well presented and analysed. Tables and figures are mostly clear and well organised. The report is interesting, but the manuscript contains a number of errors that need to be corrected before further processing of the article. Below are some points that should be considered before further processing:

·         All names used in the article such as in vitro, in vivo or in silico must be written in italics, including the title of the article!

·         All Latin names of plants throughout the article must be written botanically correct and not in italics! Correct throughout the article! For example: Plinia cauliflora (Mart.), Name  name L. etc..

·         Tab. 1 = author Adams 2007 does not have a citation number in the table, it is listed as number 90 in the list of references. Furthermore, I do not understand why Sparkman is also listed there? He is not in the bibliography at all.

·         Throughout the text, the fixed p-value should be in italics.

·     I assume that the authors have been practising on an older version of the template, because they have listed the year 2020 on the template - it is already 2023, the lines are not numbered, so it is difficult to directly mark every single error in the article, the authors will have to search!

·        Figure 6: What does the exclamation mark in the picture description mean? = ! vs indomethaci.

·     The bibliography is incorrectly numbered from number 36 onwards! Please correct according to the instructions of the journal Microorganisms!

·         In the article, I miss the highlighted novelties (add at the end of the Introduction section) and the future perspectives and limitations of this study (add in the Conclusion section).

·         Supplementary materials contains a chromatogram of the chemical composition of the essential oil, but it is very unclear, try to replace it... not all the peaks and their labels are visible!

Comments on the Quality of English Language

Moderate editing of English language is required.

Author Response

Reviewer: All names used in the article such as in vitro, in vivo or in silico must be written in italics, including the title of the article!

Our answer: Thank you for your kindly observation. We already revised all the manuscript and placed all this words in italic.

Reviewer: All Latin names of plants throughout the article must be written botanically correct and not in italics! Correct throughout the article!

For example: Plinia cauliflora (Mart.), Name name L. etc..

Our answer: Thank you for your kindly reminder. We already revised the entire manuscript and correct all scientific names for the italic format.

Reviewer: Tab. 1 = author Adams 2007 does not have a citation number

in the table, it is listed as number 90 in the list of references.

Furthermore, I do not understand why Sparkman is also listed there? He is not in the bibliography at all.

Our answer: Thank you for the kindly observation. We are sorry for the confusion. Adams 2007 was the only one to be cited. We already correct the citation on the table.

Reviewer: Throughout the text, the fixed p-value should be in italics.

Our answer: We are sorry for that. We have changed throughout the text.

Reviewer: I assume that the authors have been practising on an older version of the template, because they have listed the year 2020 on the template - it is already 2023, the lines are not numbered, so it is difficult to directly mark every single error in the article, the authors Will have to search!

Our answer: We are sorry for that. We were using the template given towards e-mail. Since your comments, we have looked up on the website, and used what we believe is the newer template.

Reviewer: Figure 6: What does the exclamation mark in the picture

description mean? = ! vs indomethaci.

Our answer: * is referred to the statistics comparing to the control group, and ! is referred to the statistics comparing to the indomethacin group.

Reviewer: The bibliography is incorrectly numbered from number 36 onwards! Please correct according to the instructions of the journal Microorganisms!

Our answer: Thank you for the kindly observation. We are sorry for the confusion. All references and citations were revised and corrected.

Reviewer: In the article, I miss the highlighted novelties add at the end of the Introduction section) and the future perspectives and limitations of this study (add in the Conclusion section).

Our answer: Thank you for your considerations to enhance the writing structure. We have upgraded the information, and have added these topics.

Reviewer: Supplementary materials contains a chromatogram of the chemical composition of the essential oil, but it is very unclear, try to replace it... not all the peaks and their labels are visible!

Our answer: Thank you for your kindly concern. But unfortunately, we cannot be replaced for a better figure since that it is how the equipment generated the figure for us.

Reviewer: Moderate editing of English language is required.

Our answer: We apologize. Our text has been double checked, and we hope that the writing is better in this current version.

Reviewer 3 Report

Comments and Suggestions for Authors

Comments and Suggestions for Authors:

The article titled “Leishmanicidal activities of essential oil from Plinia cauliflora (Mart.) Kausel: in vivo, in vitro, and in silico approaches” is interesting.  However, the manuscript should go under refinement by adding some information mentioned below to make the results more clear and understandable:

1. Authors claim trough the manuscript that they had perform quantitative and qualitative analysis of essential oil. However, there is no support for GC-MS quantitative analysis. It is unclear did they just calculated percentages of peak areas in GC-MS, or they perform quantitative analyses with identified compound standards? If they did not perform quantitative analyses of essential oils with standards, than text through all parts of the manuscript should be changed so there would be no confusion are they representing and discussing percentages or relative percentages of identified compounds in the oils.

2. In the subtitle “Plant material “ authors did not provide GPS data of plant material collection.

Author Response

Reviewer: Authors claim trough the manuscript that they had perform quantitative and qualitative analysis of essential oil. However, there is no support for GC-MS quantitative analysis. It is unclear did they just calculated percentages of peak areas in GC-MS, or they perform quantitative analyses with identified compound standards?

If they did not perform quantitative analyses of essential oils with standards, than text through all parts of the manuscript should be changed so there would be no confusion are they representing and discussing percentages or relative percentages of identified compounds in the oils.

Our answer: Thank you for your kindly observation. Indeed, we do not performed a quantitative analysis, so we correct all the parts in the manuscript that gives a misunderstand regarding this aspect.

Reviewer: In the subtitle "Plant material " authors did not provide GPS data of plant material collection.

Our answer: Thank you for your kindly observation. We already provided this information in the respective section of the work. It is highlighted in yellow.

Round 2

Reviewer 2 Report

Comments and Suggestions for Authors

Dear Editor and authors,

The authors have visibly improved and corrected the article. However, it is necessary to focus again on the English in the article and to correct all the shortcomings. Otherwise, I think that after correcting the English language and some typos, the article will be suitable for publication in the journal Microorganisms.

Comments on the Quality of English Language

Minor editing of English language is still required.

Author Response

We apologize for the inconvinient. We have gone through the text another round, and have improved the text, and reduced typos.

We believe the text has a proper structure in this version.